# Towards efficient deep spiking neural networks construction with spiking activity based pruning

## Abstract

Spiking neural networks (SNNs) drawing inspiration from the biological nervous system possess the distinctive advantage of being biologically interpretable and energy-efficient. In recent years, there has been a rise in deep and large-scale SNNs structures that exhibit high performance across various complex datasets. However, within these structures, a significant number of redundant structural units are often present, compelling the need to compress the network models of SNNs to more effectively harness their low-power advantage. Currently, most model compression techniques for SNNs are based on unstructured pruning of individual connections, which requires specific hardware support. Receptive field cells in the biological visual system have influenced a crucial concept in deep learning: convolutional kernels. Hence, we propose a structured pruning approach based on the activity levels of convolutional kernels named Spiking Channel Activity-based (SCA) network pruning framework. Inspired by synaptic plasticity mechanisms, our method dynamically adjusts the network's structure by pruning and regenerating convolutional kernels during training, enhancing the model's adaptation to the current target task. While maintaining model performance, this approach refines the network architecture, ultimately reducing computational load and accelerating the inference process. We conducted experiments on static datasets including CIFAR10, CIFAR100 and DVS-CIFAR10. Experimental results demonstrate that this method incurs only about 2% accuracy loss while retaining 20% of the channels. This indicates that structured dynamic sparse learning methods can better facilitate the application of deep SNNs in low-power and high-efficiency scenarios.

## 1 Introduction

Spiking Neural Networks (SNNs), known as the third generation of artificial neural networks (Maass, 1997), are renowned for their advantages in biological interpretability and low power consumption. Inspired by biological information processing mechanisms, unlike traditional artificial neural networks (ANNs), SNNs utilize spike sequences for feature encoding, transmission, and processing. Deployed on neuromorphic chips, SNNs capitalize on event-driven and asynchronous computation to significantly reduce computational complexity and minimize unnecessary overhead (Davies et al., 2018). It's noteworthy that the human brain operates on just 20 watts of power (Mink et al., 1981). The synaptic connection count undergoes a rapid increase and a gradual decrease throughout the lifecycle (Cizeron et al., 2020). In the process of cognition, the brain network's structure exhibits plasticity, meaning synaptic connections can strengthen, weaken, establish, or eliminate under the influence of learning and experience (Trachtenberg et al., 2002) (Stettler et al., 2006) (Holtmaat & Svoboda, 2009) (Barnes & Finnerty, 2010). However, many SNNs adopt fixed network structures from conventional deep learning (Wu et al., 2018) (Fang et al., 2021a), which may contain illogical or redundant components. Therefore, to better leverage the energy-efficient advantages of SNNs, drawing inspiration from the mechanisms of synaptic plasticity in biology, it is highly essential to adaptively learn appropriate and sparse network structures during training according to the objectives of the task.

In recent years, much research has focused on the adaptive sparse structural learning of SNNs, training a lightweight network from scratch. Kappel et al. (2015) proposed a framework of synaptic sampling

for Bayesian inference to optimize both the structure and parameters of the network. Inspired by the mechanisms of brain rewiring, Bellec et al. (2017) introduced the Deep R method for ANNs. This method trains deep networks under strict connectivity constraints. They extended the Deep R approach to RSNN and LSNN (Bellec et al., 2018) while also implementing it on the prototype chip of the 2nd generation SpiNNaker system (Liu et al., 2018). The Spike-Thrift method employs attention mechanisms for pruning and growing connections during training (Kundu et al., 2021). Chen et al. (2021) introduced the Grad R method, defining the size and density of dendritic spines as the absolute values of weights and the pruning and growth occur through adaptive competition. They further defined dynamic pruning methods for excitatory and inhibitory transformation of dendrites (Chen et al., 2022). However, most of the aforementioned approaches for SNNs structural learning focus on sparsification at the level of individual connections. This approach is non-structured and exhibits irregular and unstable characteristics, making it less suitable for hardware deployment and optimization. On the other hand, structured sparse methods are more amenable to achieving lightweight networks, reducing the complexity of memory access, and displaying superior hardware adaptability.

This paper introduces an adaptive and structured sparse learning framework for SNNs, inspired by the mechanisms of synaptic plasticity in biological neural networks. As shown in Figure 1(a), the proposed SCA framework achieves network lightweight at the granularity of convolutional kernels, akin to the sparse responses of receptive cells in the biological visual system (Houweling & Brecht, 2008) (Hu et al., 2014). During training, the SCA framework jointly learns both the network's parameters and its structure. It accomplishes this through a self-discovery process of pruning and regrowing, guided by the structure learning rule, leading to the identification of a lightweight yet high-performance network model. The structure learning rule determines the importance of channels based on the spiking activity level, to identify the guidance for dynamic topology updates. Coarse-grained sparsification facilitates the learning of regularized structures, effectively reducing storage and computational costs, while also enhancing hardware compatibility. The following are the main contributions of this paper:

- The proposed SNNs sparsification framework adapts the structure and parameters of a joint learning network by using convolutional kernels as the fundamental units. Structured pruning aligns better with the characteristics of sparsified responses in the biological visual neural system, making it easier to achieve more compressed and lightweight network models.

- The structural learning rule of this framework adheres to the mechanisms of synaptic plasticity, allowing for the removal of redundancies and regeneration of certain convolutional kernels based on spiking activity level. An adaptive structural learning approach is advantageous for acquiring a network structure that is more appropriate for the target task.

- This framework is a general-purpose framework that can be applied to most directly trained deep convolutional SNNs. Experimental results demonstrate that our approach performs well on CIFAR10, CIFAR100 and DVS-CIFAR10, maintaining network performance while reducing the parameter count.

## 2    RELATED WORK

**Training Method.** The development of SNNs training methods has undergone a progression from shallow to deep architectures. The earliest SNN algorithms are primarily based on unsupervised learning principles of synaptic plasticity. These methods often utilize fully connected single-layer network structures such as STDP (Song et al., 2000), Oja (Oja, 1982), and BCM (Bienenstock et al., 1982), or shallow convolutional architectures like SDNN (Kheradpisheh et al., 2018), SpiCNN (Lee et al., 2018), and ReStoCNet (Srinivasan & Roy, 2019). Typically, these approaches exhibit limited performance and are applied to less complex datasets. In recent years, numerous supervised training methods for deep SNNs have emerged, drawing inspiration from the concepts of backpropagation. These methods significantly enhance the performance of SNNs on intricate datasets. The STBP method employs a gradient surrogate approach to implement error backpropagation across both temporal and spatial dimensions (Wu et al., 2018). This technique can be applied to sophisticated network architectures like VGG and ResNet. Building upon this, Zheng et al. (2021) have introduced the tdBN method, which offers improved feature normalization in both temporal and spatial dimensions. Fang et al. (2021b) introduced the PLIF model that enables the updating of membrane

time constants during the training process. Deng et al. (2022) proposed the TET method to address momentum loss during the surrogate gradient descent process. SEW-Resnet (Fang et al., 2021a) and MS-ResNet (Hu et al., 2021) adjust the structure of residual networks to achieve better identity mapping of residual blocks. These addresses challenges related to gradient explosion or vanishing and enables the successful training of SNN models with over a hundred layers. These techniques hold promise for the creation of intricate, large-scale deep SNNs with impressive performance.

**Structural Learning Method.** These deep SNNs often come with a large number of parameters and significant computational costs, making the pruning of redundant connections essential. Pruning methods can generally be categorized as non-structured and structured. Non-structured pruning typically involves removing individual connections to reduce the parameter count. Deng et al. (2021) combined STBP with the ADMM (Alternating Direction Method of Multipliers) optimization tool to achieve a sparse network structure through alternating optimization. Yin et al. (2021) devised binary "gates" to control the presence of connections, optimizing binary Bernoulli gate values through a smoothed objective function. Drawing inspiration from the dynamic changes in brain connectivity, Chen et al. (2021) proposed the Grad R method, where pruning is accompanied by the growth of certain connections in an adaptive competitive process. Chen et al. (2022) further defined a process of transformation between filamentous pseudopodia and mature dendritic spines, allowing connections to be adaptively pruned or reactivated. The ESL-SNNs method initializes a sparse network structure using the Erdos-Renyi random graph approach and progressively adapts the network structure from scratch (Shen et al., 2023). However, Non-structured pruning, being a fine-grained method, often requires specific hardware support. On the other hand, structured pruning typically operates at the channel or layer level, yielding more compact network structures. Chowdhury et al. (2021) employed principal component analysis to compute inter-channel correlations, identifying redundant channels within SNNs. This method is primarily inspired by techniques from the field of ANNs and applies to SNNs.

## 3 PRELIMINARY

**Spiking Neuron Model.** The spiking neuron model is the fundamental unit of a spiking neural network, simulating the behavior of biological neurons in transmitting information through spikes. The mechanism underlying the generation of action potentials in biological neurons involves processes of depolarization, overshoot, and repolarization. The spiking neuron model simulates the mechanism of biological action potentials and thus establishes a simplified mathematical model, which consists of three equations: charging, discharging, and resetting. As shown in the following Eq. (1), the spiking neuron accumulates the input stimuli $X_t$ from presynaptic neurons, integrating all the currents received from these neurons. If the membrane potential $H_t$ of the spiking neuron exceeds a certain threshold $V_{\text{th}}$, it will fire a spike and reset to the resting potential $V_{\text{reset}}$.

$$
\begin{aligned}
H_t &= f(V_t - 1, X_t) \\
S_t &= g\left(H_t - V_{\text{th}}\right) = \Theta\left(H_t - V_{\text{th}}\right) \\
V_t &= H_t \cdot (1 - S_t) + V_{\text{reset}} \cdot S_t
\end{aligned}
\tag{1}
$$

Where $\Theta(\cdot)$ denotes the Heaviside step function, thus $s_t$ is 1 if a spike is fired, and 0 otherwise. The function $f(\cdot)$ is the equation describing the dynamics of spiking neurons. In this paper, the spiking neuron models we use are the Integrate-and-Fire (IF) neuron model and the Leaky Integrate-and-Fire (LIF) neuron model as presented in Eq. (2) of Spikingjelly (Fang et al., 2020).

$$
\begin{aligned}
V_t &= f(V_{t-1}, X_t) = V_{t-1} + X_t \\
V_t &= f(V_{t-1}, X_t) = V_{t-1} + \frac{1}{\tau_m}\left(-\left(V_{t-1} - V_{\text{reset}}\right) + X_t\right)
\end{aligned}
\tag{2}
$$

Where $\tau_m$ represents the time constant of membrane potential.

**Surrogate Gradient.** Differing from conventional neural networks, the spiking neurons in SNNs transmit feature information through a sequence of discrete spikes. As depicted in Eq. (3), the backpropagation in SNNs leverages the chain rule to compute gradients across spatial and temporal dimensions, akin to the Backpropagation Through Time (BPTT) algorithm in Recurrent Neural Networks (RNNs).

$$
\frac{\partial L}{\partial W} = \sum_{t=1}^{T} \frac{\partial L}{\partial H_t} \frac{\partial H_t}{\partial W} = \sum_{t=1}^{T} \frac{\partial L}{\partial S_t} \frac{\partial S_t}{\partial H_t} \frac{\partial H_t}{\partial X_t} \frac{\partial X_t}{\partial W}
\tag{3}
$$

Due to the binary nature of spikes ($S_t \in \{0, 1\}$), $\frac{\partial S_t}{\partial H_t}$ is non-differentiable. Hence, to enable the utilization of backpropagation for the direct training of SNNs, the gradient surrogate method employs differentiable functions $g(x)$ to substitute for the non-differentiable spikes $\Theta(x)$. The gradient surrogate function utilized here is the sigmoid function $g(x) = \text{Sigmoid}(\alpha x) = \frac{1}{1+e^{-\alpha x}}$, where $\alpha$ is used to control the smoothness of the function.

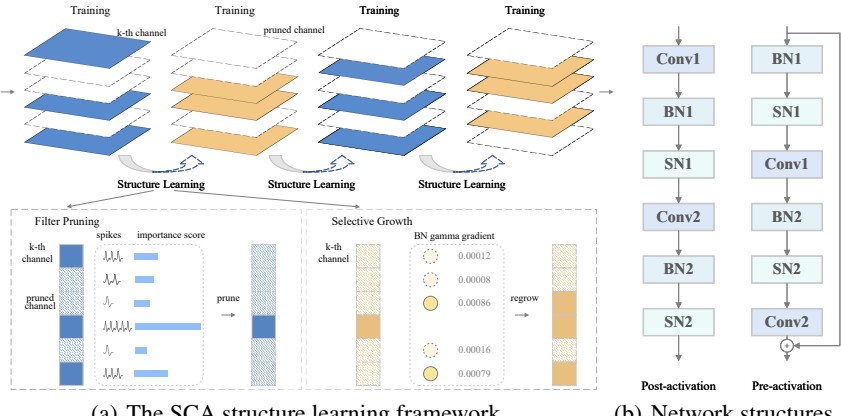

(a) The SCA structure learning framework.   (b) Network structures.

Figure 1: The schematic illustration of the SCA structure learning framework. (a) The proposed framework follows structure learning rules to adaptively learn lightweight network models during training. (b) The proposed framework is applicable to various network structures like pre-activation, post-activation, etc.

## 4 METHODOLOGY

This section introduces the dynamic structural pruning framework proposed in this paper, SCA network pruning method. This method determines the importance of channels based on their spiking activity and dynamically optimizes the network structure during the training process. It is applicable to various convolutional network architectures, including VGG, ResNet, and others.

### 4.1 THE SCA STRUCTURE LEARNING FRAMEWORK

As depicted in Algorithm 1, the SCA structure learning method simultaneously performs joint learning of both the network's weight parameters and its topological structure during the training process. Its inspiration originates from the plasticity of the biological neural system, resembling how biological neural networks optimize information transmission through processes of synaptic synaptic modulation, establishment and elimination. In the structural learning phase, the significance of each channel is determined by assessing the spiking activity level. This method identifies redundant channels with lower spiking activity and removes them while generating a subset of channels in each or several iterations. The network's structural changes are recorded using a mask. During the weight learning phase, the network's weights are updated using gradient surrogate method. Then, repeat the two aforementioned processes until the model converges. During the training process, we added L1 regularization to better identify redundant structures. Finally, a new model is obtained by completely removing redundant channels according to the mask, ensuring a meticulously optimized lightweight network without compromising its performance.

### 4.2 CHANNEL IMPORTANCE SCORE

In SNNs, features were encoded and transmitted in the form of spike sequences. We obtain importance assessment scores by calculating the spiking neuron activity of each channel in the overall network model's feature map. In biological neural cells, the resting membrane potential is around -65mV. When excited stimuli are received, the membrane potential increases, leading to a decrease in polarization level; this process is termed depolarization. On the other hand, inhibitory stimuli result

---

**Algorithm 1** The Overall Training Framework of SCA Structure Learning Framework.

---

**Input:** Input data $X$. Labels $Y$.
**Output:** Pruning channels ratio $p\%$. Mask update ratio $q\%$. The weight $W$. The weight mask $M$.
 1: Initialize the weight $W$;
 2: **for** $i$ in $[1, epoch]$ **do**
 3:     Learn weight parameters based on surrogate gradient under L1 regularization;
 4:     Learn weight connection based on structure learning rule:
 5:     (1) Prune $q\%$ channels, resulting in the removal of $(p + q)\%$ channels;
 6:     (2) Regrow $q\%$ channels, maintaining a pruning ratio of $p\%$;
 7:     **for** each layer of the model **do**
 8:         $W = M \odot W$;
 9:     **end for**
10: **end for**
11: Completely remove the channels corresponding to zero positions in the mask to obtain the compressed network model.
12: **return:** The lightweight SNN.

---

in a decrease in membrane potential, causing an increase in polarization level; this process is referred to as hyperpolarization (Gerstner et al., 2014). Drawing inspiration from this biological mechanism, neurons with lower polarization levels can be considered less important, and their removal has a relatively minor impact on the network's accuracy. We use $r_k^l$ to denote the k-th channel of the l-th layer in the network. In each iteration, the average membrane potential of this channel on the training set is computed, as shown in Eq. (4).

$$r_k^l = \frac{1}{N} \frac{1}{T} \left( \sum_{n=1}^{N} \sum_{t=1}^{T} \left\| H_k^l(t) \right\| \right) \tag{4}$$

Where $N$ represents the number of samples in the training set, and $T$ is the time step of the SNN. $\left\| H_k^l(t) \right\|$ denotes the L1 norm of the membrane potential of that channel's feature map. In the feature map, the membrane potential of a specific neuron at time $t$ is denoted as $H_t^{ij}$ ($\left\| H_k^l(t) \right\| = \left[ H_t^{ij} \right]_{h \times w}$).

If $H_t^{ij}$ is positive, it is referred to as an excitatory postsynaptic potential (EPSP); otherwise, it is an inhibitory postsynaptic potential (IPSP). In other words, the degree to which $H^{ij}t$ deviates from 0 is used to determine the activity level of the spiking neuron, i.e. Spiking Channel Activity, similar to the polarization level of biological cells (Gerstner et al., 2014).

### 4.3 STRUCTURE LEARNING RULE

**Filter Pruning.** In order to eliminate redundant portions in the network that contribute minimally to the target task or cause interference, a certain proportion of convolutional kernels are pruned based on channel importance scores. Initially, the sparsity ratio is set to $p\%$. We first calculate importance scores for all channels in the network and sort these scores, considering channels with lower scores as redundant components. This approach enables a more balanced reduction of redundancy across the entire network, leading to greater compression benefits while maintaining performance. We employ a mask to record the network's structure, where 0 denotes pruned channels and 1 represents retained ones. To better explore optimal network structures, during each iteration, a further $q\%$ of pruning convolutional kernels are introduced. Simultaneously, $q\%$ of the pruned convolutional kernels are reselected for regeneration, maintaining a consistent network sparsity ratio $p\%$.

**Selective Growth.** During the phase of selective growth, we determine the channels to be reactivated based on the gradient magnitudes of the Gamma parameters $\gamma$ within the Batch Normalization (BN) layer. The Gamma parameters in the BN layers, acting as scaling factors, influence the range of features in the output. When pruned channels exhibit significant gradients in their Gamma parameters, it suggests the potential for these channels to regain their activity. Consequently, we can selectively regrow the network's channels with higher Gamma parameter gradients, aiming to optimize the network structure. After structural adjustments, the network is iteratively trained for one or several epochs, with the channels masked with 0 participating in the training. Through the process of pruning and regrowth, this framework mitigates the issue of potentially irreversible losses caused by pruning. Moreover, it facilitates the correction of accuracy losses resulting from erroneous pruning by allowing reactivation through regrowth.

## 4.4 VARIOUS NETWORK ARCHITECTURES

This framework can be applied to various convolutional spiking neural network architectures. As shown in Figure 1(b), the SN denotes the spiking neuron layer. For post-activation network structures like VGG and ResNet, we employ the process of pruning and regeneration using the BN layer and spiking neuron layer immediately following the conv layer. For pre-activation network structures like PreResNet, in the architecture on the right side, the BN1 and SN1 of the current block are used to determine filter pruning for the convolutional layer Conv2 of the previous block, while BN2 and SN2 are used to decide that of Conv1 for the current block. We will elaborate on the structural details employed in the experimental section.

## 5 EXPERIMENTS

### 5.1 EXPERIMENT SETTINGS.

Our experiments are conducted using the SpikeJelly framework (Fang et al., 2020), an SNN framework implemented based on PyTorch. We perform experiments on both static datasets and neuromorphic datasets. CIFAR-10 and CIFAR-100 are commonly used static image classification datasets, containing 10 and 100 classes, respectively. The images in these datasets are 32x32 RGB images. To input images into SNNs, encoding is required, and we use a time step of 4. The DVS-CIFAR10 dataset is a neuromorphic dataset, and we divide it into training and testing sets with 9000 and 1000 samples, respectively. Event data needs to be integrated into frame data, and we use 20 time steps for this purpose. The network structures used in the experiments are based on VGG and Pre-ResNet architectures. For the DVS-CIFAR10 experiments, the network structure is 64C3-AP2-128C3-AP2-128C3-AP2-256C3-AP2-256C3-AP2-10FC. The number of iterations for all experiments is set to 300 epochs.

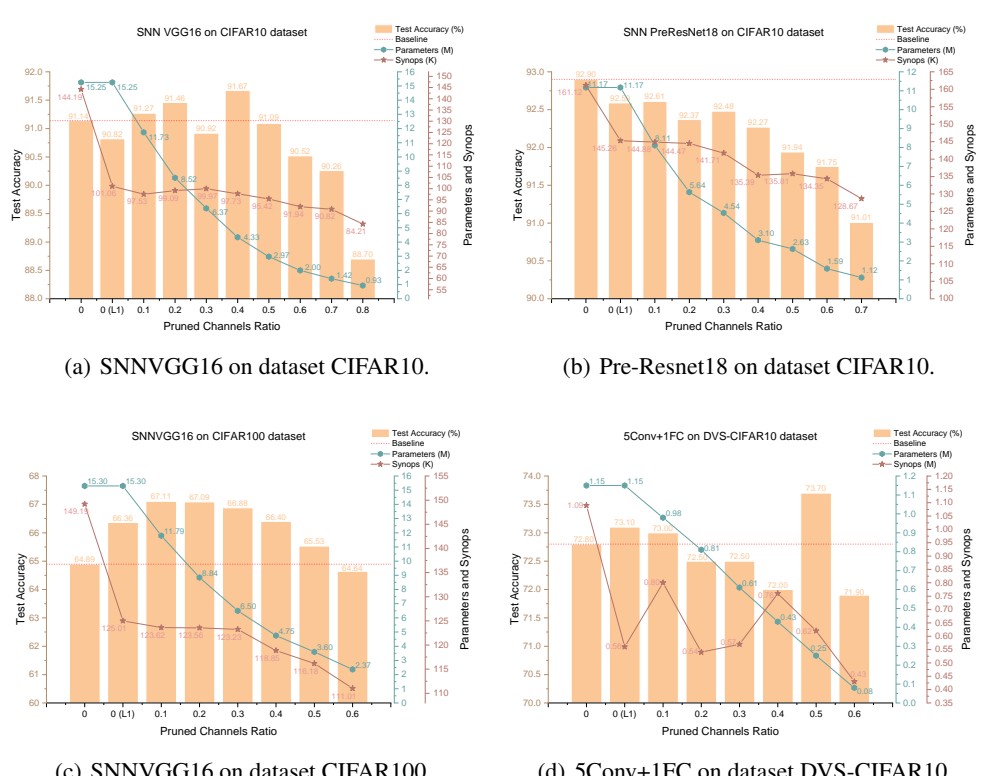

(a) SNNVGG16 on dataset CIFAR10.

(b) Pre-Resnet18 on dataset CIFAR10.

(c) SNNVGG16 on dataset CIFAR100.

(d) 5Conv+1FC on dataset DVS-CIFAR10.

Figure 2: The performance of the SCA structure learning framework.

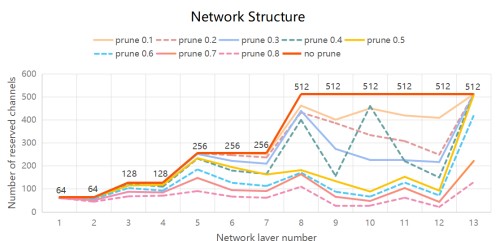
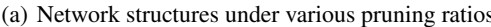
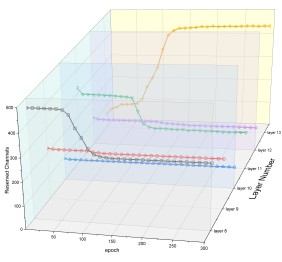

(a) Network structures under various pruning ratios.

(b) The evolution of network structure during the training process.

Figure 3: Analysis of structural changes in the SCA framework.

## 5.2 EVALUATION ON DIFFERENT DATASETS.

As shown in Figure 2, we present the experimental results of our framework on various datasets. We set different channel pruning ratios and maintain a fixed level of sparsity during the training process. After training is completed, we completely remove redundant channels and construct a new lightweight model for evaluation. The computational energy consumption of the network is evaluated using the number of parameters and synaptic operations.

**Performance analysis.** The bar chart in Figure 2 represents the model's accuracy on the test set under different channel pruning ratios. The green and red lines indicate the final model's parameters and synaptic operations, measured in units of M ($10^6$) and K ($10^3$). It can be observed that as the channel pruning ratio increases, the model's test accuracy experiences only a minor loss, while the number of parameters decreases significantly, and in some cases, even at lower pruning ratios, the accuracy slightly improves. This indicates that our framework can compress the network's parameter count to obtain a lightweight model while maintaining close-to-original high performance. For CIFAR-10, in the case of the VGG16 model shown in Figure 2(a), when the parameter count is compressed to approximately $10\%$ of the original size (i.e., 1.42 million parameters), the accuracy drops by less than $1\%$. When the parameter count is compressed to around $5\%$ (i.e., 0.93 million parameters), the accuracy decreases by about $2\%$. When the parameter count is reduced by approximately $30\%$, the model's accuracy reaches $91.67\%$, showing an increase of $0.53\%$. For the ResNet model depicted in Figure 2(b), when the parameter count is compressed to approximately $20\%$, the accuracy loss is less than $1\%$. For CIFAR-100 in Figure 2(c), when the model's parameter count is compressed to $20\%$ of the original (i.e., 3.60 million parameters), the accuracy still increases by $+0.64\%$. In the case of DVS-CIFAR10, as shown in Figure 2(d), even when the parameter count is reduced to only $6.95\%$ of the original (i.e., 0.08 million parameters), the accuracy decreases by less than $1\%$. Furthermore, when the parameter count is compressed to approximately $20\%$ of the original (i.e., 0.25 million parameters), the accuracy improves by nearly $1\%$.

**Energy Consumption Analysis.** The synaptic operations (SynOps) of SNNs can represent the network's complexity and computational cost. As seen from the red line in Figure 2, with an increase in the network's channel pruning ratio, the synaptic operations gradually decrease. As shown in the Figure 2(a), for SNNVGG16 on the CIFAR-10 dataset, when the redundant channel removal ratio reaches 0.8, nearly half of the synaptic operations can be reduced. For DVS-CIFAR10 in Figure 2(d), when the number of synaptic operations (0.43M) is less than half of the original, the performance only decreases by less than $1\%$. This indicates that our method can reduce computational costs, improve resource utilization, and enhance network performance, which is highly beneficial for deploying SNNs in resource-constrained environments or achieving more efficient neural computations.

## 5.3 ANALYSIS OF NETWORK STRUCTURE LEARNING.

**Structure Analysis.** In order to better analyze the learning process of the network structure, we visualized the changes in the number of channels for each layer of the network, as shown in Figure 3. In Figure 3(a), different lines represent the number of channels in each layer of the final network structure under different pruning ratios, using SNNVGG16 on the CFAR10 dataset as an example.

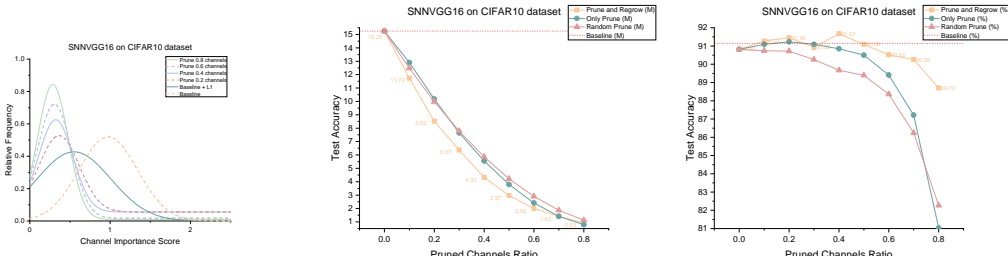

(a) Channel importance score distribution.

(b) The accuracy of ablation experiments.

(c) The parameters of ablation experiments.

Figure 4: Analysis of the effectiveness of the SCA structure learning framework.

It can be observed that there is a certain pattern in the change of channel numbers under different pruning ratios. The removal of more channels in the deeper layers suggests their lower importance, while channels in the shallower layers appear to be relatively more important. In deep network structures, deeper layers extract more abstract and advanced features, but as the layers get deeper, they become harder to train, leading to the presence of many redundant connections. In Figure 3(b), the changes in the number of channels for convolutional layers 8-13 of the SNNVGG16 model under a pruning ratio of 0.5 during the training process are visualized. It can be seen that as the network converges, the model's structure stabilizes to some extent. This indicates that the structural learning framework, as the network structure evolves, autonomously adapts to an appropriate structure.

**Channel Importance Score Distribution.** As shown in Figure 4(a), we visualized the distribution of channel importance scores for the entire network at different pruning ratios. The points where the importance score is 0 represent the removed channels that are no longer firing. It can be seen that the original network's importance score distribution has a higher mean and is more scattered, indicating that the channels have higher spiking activity levels and larger variance. As the pruning ratio increases, the overall network activity decreases and tends towards zero, indicating that removing redundant channels reduces the overall energy consumption of the network.

## 5.4 ANALYSIS OF ABLATION EXPERIMENTS.

To further demonstrate the effectiveness of the structural learning method in this framework, we conducted ablation experiments, as shown in Figure 4(b) Figure 4(c). The yellow curve represents the testing accuracy and parameter count of the SNNVGG16 method on the CIFAR10 dataset. The green and red curves represent the results of pruning only without regrowth and random pruning at initialization, respectively. It can be observed that the testing accuracy follows the order 'Prune and Regrow' > 'Only Prune' > 'Random Prune', and the parameter count for the latter two is higher than that of 'Prune and Regrow.' When the pruning ratio is 0.8, our framework achieves only about a 2% decrease in accuracy, while the accuracy of only pruning and random pruning drops significantly. When only pruning is performed, after an initial mask update at the beginning of training, the mask hardly gets updated during subsequent training, indicating that the regrowth process can reactivate mistakenly pruned channels. Random pruning, which prunes with the same probability for each layer, cannot accurately identify redundant structural units. Therefore, the SCA framework's mechanism for identifying redundant channels and reactivating pruned channels autonomously searches for an appropriate network structure.

## 5.5 COMPARISON WITH OTHER METHODS.

As shown in Table 1, we compared our method with existing SNNs pruning methods, presenting the loss in test accuracy for different pruning ratios and the percentage of remaining parameters. On the CIFAR-10 dataset, at approximately 9.31% of the original parameter count, our method achieved a less than 1% decrease in accuracy. This performance is similar to the PCA-based channel pruning method (Chowdhury et al., 2021). However, the PCA-based method use a time step of 25, much larger than our method's 4. Compared to the SD-SNN method (Han et al., 2022), our method also exhibited

advantages in parameter compression and performance loss. The SCA-based method achieved a 0.53% improvement in accuracy at only 28.39% of the original parameter count, outperforming the ADMM-based (Deng et al., 2021) and ESL-SNNs (Shen et al., 2023) methods. The SCA-based method may not excel in parameter compression compared to Attention-based (Kundu et al., 2021), Grad R (Chen et al., 2021), and Dendritic-based (Chen et al., 2022) methods. However, our approach employs structured pruning specifically targeting convolutional layers and efficiently retrains from scratch in a more energy-efficient manner. Ultimately, it allows for the complete removal of redundant structural units, resulting in a new model with improved hardware-friendliness. On the CIFAR100 dataset, the accuracy loss is 0.25% when the parameters are 16.47% of the original, which is better than the the PCA-based method (Chowdhury et al., 2021). The attention-based method (Kundu et al., 2021) requires a process involving ANN-SNN conversion, while our method is comparatively more energy-efficient. On the DVS-CIFAR10 dataset, there was a slight improvement in accuracy when compressing fewer parameters. Additionally, the network structure used was simpler compared to ESL-SNNs (Shen et al., 2023). Therefore, the experimental results indicate that our SCA structured pruning method can efficiently identify redundant structural units and adapt the lightweight network autonomously for the target task through training from scratch. This method results in a thoroughly lightweight model with a more regular structure, making it easier to deploy on hardware chips compared to unstructured pruning.

Table 1: Comparison of Experimental Performance with Other Methods.

| Dataset | Method | Network Architecture | Granularity | ACC. (%) | ACC. Loss (%) | Connectivity(%) |
|---|---|---|---|---|---|---|
| CIFAR10 | Attention-based (Kundu et al., 2021) | VGG16 | Weight | 91.13 | -0.39 -0.98 | 5.00 2.99 |
| | ADMM-based (Deng et al., 2021) | 7Conv+2FC | Weight | 89.53 | -2.16 -3.85 | 25.00 10.00 |
| | Grad R (Chen et al., 2021) | 6Conv+2FC | Weight | 92.84 | -0.30 -0.34 -0.81 | 28.41 12.04 5.08 |
| | Dendritic-based (Chen et al., 2022) | 6Conv+2FC | Weight | 92.84 | -0.35 -2.63 | 2.23 0.75 |
| | ESL-SNNs (Shen et al., 2023) | Resnet19 | Weight | 91.09 | -1.70 | 50.00 |
| | SD-SNN (Han et al., 2022) | 6Conv+2FC | Channel | 94.74 | -0.64 | 62.56 |
| | PAC-based (Chowdhury et al., 2021) | VGG9 | Channel | 90.10 | -1.06 | 7.00 |
| | **SCA-based** | VGG16 | Channel | 91.14 | **+0.32 +0.53 -0.88** | **55.86 28.39 9.31** |
| CIFAR100 | Attention-based (Kundu et al., 2021) | VGG16 | Weight | 64.69 | -0.03 | 10.00 |
| | PAC-based (Chowdhury et al., 2021) | VGG11 | Channel | 68.10 | -1.70 | 10.70 |
| | **SCA-based** | VGG16 | Channel | 64.89 | **+0.64 -0.25** | **23.52 16.47** |
| DVS-CIFAR10 | ESL-SNNs (Shen et al., 2023) | VGG8 | Weight | 78.30 | -0.28 | 10.00 |
| | **SCA-based** | 5Conv+1FC | Channel | 72.80 | **+0.90 -0.90** | **21.73 6.95** |

# 6 CONCLUSION

Lightweight and high-performance SNNs can better leverage their advantages of low power consumption. The use of structured pruning methods can result in regular, sparse SNN models, making them more hardware-friendly. The approach proposed in this paper starts from the perspective of biological plasticity, combining pruning and regrowth in an adaptive manner during training to explore suitable lightweight network structures. This approach allows for the compression of network parameters and inference computations while maintaining network high performance. This is of significant value for deploying high-performance, low-memory SNNs on neuromorphic chips.

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
