# OpenReview forum: "Towards efficient deep spiking neural networks construction with spiking activity based pruning"
_ICLR.cc/2024/Conference — ICLR 2024 Conference Withdrawn Submission_

### Official Review · Reviewer_AYqQ · 2023-10-30

**Soundness:** 2 fair
**Presentation:** 3 good
**Contribution:** 2 fair
**Rating:** 3
**Confidence:** 4

**Summary:**

This paper delves into optimizing Deep Spiking Neural Networks (DSNNs). Recognizing that traditional methods for training and constructing DSNNs face challenges in efficiency, the authors introduce an innovative approach using "Spiking Activity Based Pruning." This method aims to prune insignificant neurons based on their spiking activity, thereby reducing network complexity without compromising performance. This paper also combine regrowth with pruning together, offering a potential avenue for making DSNNs more streamlined and efficient for real-world applications

**Strengths:**

1.	The work is interesting by including both pruning and regeneration together, and proved the effectiveness of this algorithm on several datasets.
2.	The pruning is on a new dimension: the channels.
3.	The paper is well organized and well-written.

**Weaknesses:**

1. The algorithm demonstrates efficacy on both CIFAR10 and CIFAR100 datasets; however, it has not been assessed on larger datasets like ImageNet.

2. The term "redundant" is frequently used in the paper to describe neurons with a lower likelihood of firing. However, the usage of "redundant" warrants careful consideration because a) even though certain neurons might fire infrequently, they could possess distinct functionalities in specific contexts, and b) if two high-firing neurons exhibit identical firing patterns, they could be considered redundant.

3. The rationale behind drawing inspiration from the definitions of depolarization and hyperpolarization for this study appears somewhat tenuous. The paper lacks a clear explanation of plasticity that aligns with the proposed method. If we were to interpret the membrane potential as a hidden layer of firing rates, then the concept of plasticity could be analogous to the principle that reduced firing correlates with decreased wiring, reminiscent of Hebb's rule. Alternatively, from a different perspective, the pruning might be seen as an L1 regularization applied to hidden values (the membrane potential) rather than directly on firing rates. This raises the question: is the novelty of this research rooted in the integration of pruning with regrowth?

**Questions:**

See the weaknesses.

1. Does this algorithm work on large dataset like ImageNet
2. Which part: the regrowth part, or the prune part results in the effectiveness of this algorithm
3. Please revise the description of "redundant"

---

### Official Review · Reviewer_vHij · 2023-10-31

**Soundness:** 3 good
**Presentation:** 4 excellent
**Contribution:** 2 fair
**Rating:** 5
**Confidence:** 5

**Summary:**

This paper proposes the Spiking Channel Activity-based (SCA) network pruning based on spiking activity/polarization level. It utilizes the L1 norm of the membrane potential as importance scores for channels, incorporating both pruning and selective growth of channel masks. SCA manages to achieve competitive results among other pruning algorithms of SNNs. The authors also provide results on the different affinity to sparsity across layers and the distribution of importance scores. An ablation study substantiates that the 'prune and regrow' pipeline mitigate the performance loss.

**Strengths:**

The elaboration on the SCA framework is quite clear and easy to follow. The idea of wiping out low-activity neurons is very straightforward and accords with the intuition of SNN compression since lower firing activity naturally indicates fewer influences on the post-synaptic neurons. Therefore those neurons could be safely removed from network dynamics.

**Weaknesses:**

There are still some flaws in both empirical studies and analysis.

1. Membrane potential near zero is not a necessary condition for low firing activity. The neurons could also experience continuous repolarization/hyperpolarization and maintain a negative potential with a large magnitude (L1 norm), which makes them silent and can thereby be removed without influencing the input of post-synaptic neurons. The application of the L1 norm on membrane potential seems to suggest they assume only neurons with potential near zero are insignificant, which for me is not heuristically comprehensive.
2. The *selective growth* part shares insights with RigL [1], which employs gradient as a criterion for recovering pruned weights. Proper discussions/citations should be made in these parts.
3. The DVS-CIFAR10 experiments shown in Table 1 do not follow the same network architecture as the counterpart methods. It's hard to directly compare results across different architectures since they could possess a far distinct potential for compression from each other even when the number of parameters is close.

[1] Utku Evci, et al. "Rigging the lottery: Making all tickets winners." International Conference on Machine Learning. 2020.

**Questions:**

1. The authors should first provide a convincing standard while comparing results. For example, they mention that the PCA-based method uses a time step of 25, much larger than SCA’s 4. However, these are training parameters and the authors could simply conduct pruning experiments with the same timestep of 25 to maintain a fair comparison. Larger training timesteps only suggest a probably higher performance for dense SNNs, which cannot serve as a clue for comparing pruning algorithms.
2. Why do authors believe structural pruning is essential for energy-efficient deployment on neuromorphic hardware? Event-based chips could behave completely differently in computation and storage patterns from GPU or CPU. The authors should provide more firm evidence for the necessity of exploring structural pruning in SNNs.

---

### Official Review · Reviewer_JMPU · 2023-10-31

**Soundness:** 3 good
**Presentation:** 3 good
**Contribution:** 2 fair
**Rating:** 3
**Confidence:** 4

**Summary:**

The paper introduces the Spiking Channel Activity-based (SCA) pruning framework.This method dynamically prunes and regenerates convolutional kernels during training, streamlining the network. Tests on datasets like CIFAR10 and CIFAR100 showed a minor accuracy loss while reducing 80% of the channels.

**Strengths:**

The results are promising as the model shows good performance with a much smaller number of connections.

**Weaknesses:**

Though the paper introduces some new method for pruning, I have some key concerns (also highlighted in Questions below):

1. First, the paper uses the concept of activity-based pruning, which is quite standard in deep neural networks, hence I was not able to find the key novelty of the paper as it seemed to be simply transferring DNN-based methods to SNNs. The notion of using 'channels' instead of weights is also not novel as illustrated in Table 1

2. The Methodology is purely discussed. Important details of the pruning algorithm are missing/hard to find. (Details in Questions)

Minor points:
1. Some of the figures are hard to read and follow (E.g. Fig 3, Fig 4)
2. The axes of Fig 4 (b) and (c) I think are wrong - otherwise this paper does not make much sense

**Questions:**

I have some key questions for the paper:

1. The authors say that
> the SCA structure learning method simultaneously performs joint learning of both the network’s weight parameters and its topological structure during the training process.

It would be great if the authors could add more details of how this is exactly done.

2. The pruning is done based on the hypothesis that the less active channels are unimportant. It would be interesting if the authors could quantify the information/shapley values of the channels based on the performance to support the hypothesis

3. One key concern for structured pruning in SNNs is the stability of the final model. It would be good if the authors could show the change in performance with input/parameter noise with different pruned networks

4. The authors say
> Drawing inspiration from this biological mechanism,
neurons with lower polarization levels can be considered less important, and their removal has a
relatively minor impact on the network’s accuracy.

It would be good if the authors could give more intuition on this statement

5. I am also not able to understand Fig 3(b) - why do only three of them change (black, green and yellow) and the other three do not change at all with epochs?

6. [Minor] The structure of the experimental sections seems haphazard. For example, personally, I feel it would be easier to understand if you could breakup Fig 2 into two parts to just keep Fig 2(c) in the appendix and make all the results just for CIFAR10; Fig 4(a) legends are not readable etc

7. [Minor] None of the figures are particularly easy to read - it would be great if the authors could increase the linewidth, make the legends larger

---

### Official Review · Reviewer_zKkP · 2023-11-06

**Soundness:** 2 fair
**Presentation:** 2 fair
**Contribution:** 2 fair
**Rating:** 5
**Confidence:** 4

**Summary:**

This paper proposes a structured pruning approach based on the activity levels of convolutional kernels to adjust the spiking neural network’s structure into structured sparse. The method consists of pruning and regenerating convolutional kernels during training based on the channel importance score and gradient magnitudes in batch normalization, respectively. The resulted network is proved to maintain a small loss with only 20% of the channels in several benchmark classification tasks.

**Strengths:**

-	The paper is well-written, with very clear illustration on the motivations, methods and implementations.
-	The paper proposes a very useful challenge in real case – how to structurally prune a spiking neural network, and ends up with a simple but effective solution.

**Weaknesses:**

-	The experiments are only implemented on small datasets (CIFAR10,100, and DVS-CIFAR), moreover, the network architecture that is used in the experiment originally does not have a high performance (e.g., for CIFAR10, current SOTA SNNs go to at least 94% accuracy with 4-6 time steps while CIFAR 100 is around 73%, ref [1]). These results arise the question on how effective and generalized of this method to different architectures and tasks.
-	Although the challenge proposed in this paper is quite interesting, the solution is somewhat moving the conventional solution in CNN to a spiking setting, which limits the novelty of the current paper.

Ref:

[1] Adaptive Smoothing Gradient Learning for Spiking Neural Networks, ICML23

**Questions:**

Please refer to above weakness part. In addition,

- In the Sec 4.2, the channel importance score is calculated by averaging the membrane potential in a channel to decide to prune this channel or not. Despite its biological interpretation, is there any difference compared with considering the feature map value in ANN?
- I’m also quite curious on whether the SNN properties (e.g. spike events, neural dynamics, etc.)  can innovate any specific solutions in such structured pruning problem? (namely rather than transferring ANN methods into SNN)